# Deciphering the West Eurasian Genetic Footprints in Ancient South India

**DOI:** 10.3390/genes14050963

**Published:** 2023-04-23

**Authors:** Bhavna Ahlawat, Lomous Kumar, Parayil John Cherian, Jagmahender Singh Sehrawat, Niraj Rai, Kumarasamy Thangaraj

**Affiliations:** 1Department of Anthropology, Panjab University, Chandigarh 160014, India; 2Birbal Sahni Institute of Palaeosciences, Lucknow 226007, India; 3CSIR—Centre for Cellular and Molecular Biology, Uppal Road, Hyderabad 500007, India; 4PAMA Institute for the Advancement of Transdisciplinary Archaeological Sciences, Pattanam Archaeological Site, Ernakulam 683522, India; 5Kerala Council for Historical Research & Director Pattanam Excavations, Thiruvananthapuram 695003, India; 6Centre for DNA Fingerprinting and Diagnostics, Uppal, Hyderabad 500007, India

**Keywords:** mitochondrial, Pattanam, haplogroup, West Eurasia, South Asia

## Abstract

Since 2006, Pattanam coastal village of the Ernakulam District in Kerala, India, has witnessed multi-disciplinary archaeological investigations in collaboration with leading research institutions across the world. The results confirm that the Pattanam site could be an integral part of the lost ancient port of Muziris, which, as per the material evidence from Pattanam and its contemporary sites, played an important role in the transoceanic exchanges between 100 BCE (Before Common Era) and 300 CE (Common Era). So far, the material evidence with direct provenance to the maritime exchanges related to ancient cultures of the Mediterranean, West Asian, Red Sea, African, and Asian regions have been identified at Pattanam. However, the genetic evidence supporting the impact of multiple cultures or their admixing is still missing for this important archaeological site of South India. Hence, in the current study, we tried to infer the genetic composition of the skeletal remains excavated from the site in a broader context of South Asian and worldwide maternal affinity. We applied the MassArray-based genotyping approach of mitochondrial makers and observed that ancient samples of Pattanam represent a mixed maternal ancestry pattern of both the West Eurasian ancestry and the South Asian ancestry. We observed a high frequency of West Eurasian haplogroups (T, JT, and HV) and South Asian-specific mitochondrial haplogroups (M2a, M3a, R5, and M6). The findings are consistent with the previously published and ongoing archaeological excavations, in which material remains from over three dozen of sites across the Indian Ocean, Red Sea, and Mediterranean littoral regions have been unearthed. This study confirms that people belonging to multiple cultural and linguistic backgrounds have migrated, probably settled, and eventually died on the South-western coast of India.

## 1. Introduction

The Pattanam (10° 09.434′ N Latitude and 76° 12.587′ E Longitude) archaeological site located on the South-western coast of the Indian subcontinent in the Ernakulam District of Kerala, India is part of the ancient port city of Muziris as mentioned in the classical Greco- Roman records or Muciri Pattinam as referred in classical Sanskrit and Tamil sources [1]. The archaeological investigations, so far, were undertaken by two organizations: the Kerala Council for Historical Research (KCHR), Thiruvananthapuram from 2007 to 2015 and by the PAMA Institute for the Advancement of Transdisciplinary Archaeological Sciences, Pattanam since 2018. The site is located at the delta region of the Periyar River, north of Ernakulam/Kochi and adjacent to the Arabian Sea. The ports other than Muziris on the South-western Coast as per the Periplus Maris Erythraei are Tindis, Nelcynda, and Becare which are yet to be identified on the ground. Periplus Maris Erythraei is one of the oldest navigational guides by an anonymous author, for sailors and merchants from the Red Sea to Indian Ocean ports and speaks of the maritime exchanges that existed as circuitous routes or mid-ocean routes [2,3]. These routes functioned as “mobile bazaars”, probably initially for the supply of essential goods and during the peak phase between 100 BCE and 300 CE for luxury items. It could have been a natural consequence that during the trans-oceanic interfaces which facilitated the exchanges of goods and technologies that multiple ethnolinguistic communities also migrated and settled in those thriving ports on the coastal belts [4,5,6]. Maximum numbers of ancient ports on the Indian Ocean coasts and littorals have been identified through a period, however, South-western coast port sites remained a mystery and was believed to be wiped out because of natural calamities such as floods and sea erosions. Thus, South-western coast remained archeologically barren until the exploration of the Pattanam site [2]. The Pattanam site finally provided the missing links in the puzzles of international exchange or maritime routes. This paper pieces together the skeletal pieces of the site to identify the different population groups and their further admixture in the then contemporary Indian population.

The Pattanam site (Figure 1A) depicts a settlement history from 1000 BCE to 500 CE. The chronology spans the initial settlement beginning with the Iron Age habitation with a commercial peak between the first century BCE and the third century CE. Accelerator mass spectrometry (AMS) Carbon-14 (C14) dating results on the charcoal samples confirm Iron Age settlement of Pattanam around 1000 BCE [7]. Other archaeological evidence shows the presence of maritime features such as port features including the wharf and urban planned architectural features. A total of 12 seasons of excavations at the Pattanam site have brought to light a broad range of finds; some of astounding proportions. The most important finds include ceramic numbering over 4.5 million; lapidary objects ranging from ballast to precious stone ornaments; objects made of iron, copper, lead, and gold- including coins and jewelry; as well as animal and botanical remains. Spices and forest goods were important luxury items exported from Muziris. There is strong evidence that proves that cotton weaving and advanced metallurgical technology was practiced at the site [8].

The high density of archaeological remains at the Pattanam archaeological site reveals a 2000-year-old polychrome and polyphonic urban culture with maritime links to regions across three continents—Asia, Africa, and Europe. Fine pottery of Indian and foreign origin was conspicuous for revealing the multi-cultural status of Muziris as a “rain forest” of cultures with an inclusive approach to life or a diverse variety of livelihoods complimenting one another positively. The Mediterranean (Roman) pottery, represented in the Amphora and Terra Sigilletta sherds constitutes the largest number of Mediterranean sherds from any of the Indian Ocean port sites reported, so far. (Figure 1B) [1,7,8]. The West Asian and South Arabian ceramics include turquoise glazed pottery, torpedo jars, and ovoid jars (Figure 1C). The major Indian fine pottery categories include the Indian rouletted ware as well as black and red ware. The assemblage of Indian rouletted wares was the largest and the first-ever reported from the Indian littorals and the South-western coast, respectively. (Figure 1D) The staggering 4.5 million red course ware Indian pottery (Figure 1E) signifies the crucial role of the Indian sub-continent in the trans-oceanic network. The chronology of these shreds ranges from 1000 BCE to 500 CE.

Ancient DNA studies were conducted on 13 randomly selected skeletal samples obtained from different seasons of Pattanam excavations. In total, seven bone samples were collected in the year 2007, one fragment in 2009, one fragment in 2010, one tooth sample in 2011, and two bone fragments with one tooth sample from 2013 seasons. The skeletal remains were in a very fragile state due to the tropical, humid, and acidic soil conditions of the Pattanam site. The early historic period of the site dates to the third century BCE to fourth century CE based on chronometric, morphometric, and stratigraphic analyses [9]. The AMS radiocarbon dating points towards the first century BCE - first century CE period. The paleoanthropological studies performed on the bones by biological anthropologists indicated the presence of flesh over the bones, further pointing to the practice of secondary burial, in which the dead are first cremated, and the bone remains later ceremoniously reburied [1]. The samples were later administered for computational analyses. The study aimed to establish the ancestry of the specimens based on the associated haplogroup of the specimens in the context of South Asian ancestry. This is one of the first genetic studies conducted on the specimens that were excavated at the Pattanam archaeological site. The study is based on mitochondrial DNA genotyping using a mass array Sequenom platform and the analysis is based on haplogroup determination [1].

## 2. Materials and methods

### 2.1. Methodology

#### 2.1.1. DNA Extraction from Skeletal Remains

Very stringent standards were used for the ancient DNA authentication. Among the thirteen samples, 12 were cranial remains and one was tooth sample. DNA could only be successfully extracted from 12 samples. All samples were independently processed in a dedicated ancient DNA laboratory at CSIR-CCMB, Hyderabad, India. In total, three independent DNA extractions were performed per sample to ensure the authenticity of the entire procedure. Utmost care was taken to avoid any form of contamination. The DNA was extracted using appropriate extraction controls.

Approximately 500 mg of each bone sample was first polished with sandpaper and then bone powder was obtained using drills. The bone powder was dissolved in 4.0 mL of extraction buffer (0.5 M EDTA pH 8.0, 0.5% SDS, and 500 µg/mL proteinase K) and incubated in a shaking incubator at 55 °C for overnight, followed by incubation at 37 °C for 12 h [10]. The QIAquick PCR Purification Kit (QIAGEN, Hilden, Germany) was used to capture the DNA fragments larger than 100 bp and smaller than 10 kb and to remove nucleotides, proteins, and salts. The columns are ideally suited for ancient DNA, as the templates are severely degraded in ancient DNA and target regions are quite small (100 to 250 bp). The extraction solution was centrifuged at 10,000× *g* for 10 min and 4.0 mL aliquots of the supernatant were transferred to a 50 mL tube and mixed with 5 volumes of QIAquick PB buffer. From this, 750 µL volume was loaded directly onto QIAquick columns and centrifuged at 12,800× *g* for 1 min. The flow-through was discarded and the process was repeated until all extracts had passed through the column. The DNA was washed with 750 µL of QIAquick PE buffer and centrifuged for 1 min. The flow-through was discarded and the DNA was then eluted from the column by loading 40 µL of elution buffer followed by centrifugation for 1 min. The single tooth sample was processed using a method described in an earlier study [11].

#### 2.1.2. Quality Control for Ancient DNA Extraction and Amplification

The ancient human bones used in this study were very fragile and old, and efforts were made to handle them with extreme care. One of the main concerns was to avoid any contamination from exogenous DNA. All steps (bone cleaning, cutting of bone samples, drilling, DNA extraction, and PCR preparation) were performed in two separate laminar airflows. Laboratory instruments and equipment used for DNA extraction were cleaned using commercial bleach and 70% alcohol, followed by at least 45 min of UV exposure. DNA extraction and PCR experiments were performed in two different dedicated chambers. Disposable body suits and gloves were used throughout the sample processing. Aerosol barrier tips and ultrapure DEPC-treated water were used for all the experiments performed in the ancient DNA lab.

#### 2.1.3. Genotyping of Mitochondrial DNA Using Mass Array Sequenom Platform

A total of 94 primers were designed for the multiplex polymerase chain reaction into four pools, with each primer pool having a distinct set of primers [10]. Designing primers was done using Sequenom’s MassARRAY Designer software (version 3.1). All oligos for PCR and iPLEX reactions were ordered unmodified, with standard purification from Integrated DNA Technologies, USA [10]. Forward and reverse PCR primers were ordered in 96-well deep-well plates and the final concentration of 240 μM concentrations was prepared in the ancient DNA lab. Collapsed PCR primers underwent further dilution after pooling to a working concentration of 1 μM each. Probes (for iPLEX extension) were ordered unmixed in 96-well deep-well plates at 250 to 450 μM and were diluted up to 100 μM. Various concentrations of DNA (100 pg–1 ng) were amplified in a 5.0 μL multiplex PCR reaction according to the manufacturer’s instructions. Multiplex PCR was performed in GeneAmp 9700 Thermal Cycler (Applied Biosystems, Foster city, USA) using 10X polymerase chain reaction (PCR) buffer, primer mix, dNTPs, and HotStarTaq (Qiagen, Hilden, Germany). The PCR conditions used for amplification were as follows: (1) An initial denaturation at 94 °C for 15 min for hot start followed by (2) 45 cycles at 94 °C for 20 s; annealing at 56 °C for 30 s; extension at 72 °C for 1 min; and (3) the last step included final extension at 72 °C for 3 min. After polymerase chain reaction (PCR) the products were treated with shrimp alkaline phosphatase (SAP, Amersham, Freiburg, Germany) to dephosphorylate the remaining dNTPs. To remove the residual dNTPs, the reaction conditions were set for 37 °C for 20 min followed by 85 °C for 30 min to inactivate SAP and finally the plate was removed from the Thermal Cycler and stored at 4 °C, until further use. After SAP treatment, a multiplex iPLEX reaction was performed. The iPLEX reaction components were 10X iPLEX Buffer, iPLEX Termination Mix, Primer Mix, and iPLEX Enzyme. The iPLEX reaction conditions were 94 °C for 30 s followed by 40 cycles at 94 °C for 5 s [5 cycles (52 °C for 5 s, 80 °C for 5 s)] and a final extension at 72 °C for 3 min, followed by incubation at 4 °C until the plate was removed from the Thermal Cycler. The iPLEX reaction mixtures were then purified by adding 6 mg of SpectroCLEAN cationic resin (SEQUENOM, San Diego, CA, USA) using a dimple plate and 16 μL of water. To desalt the iPLEX solution, the plate was then sealed and placed on a rotating shaker for 20 min. The finished iPLEX reaction products were spotted onto a 384-element matrix array silicon chip (Sequenom SpectroCHIP, SEQUENOM, San Diego, CA, USA) using the MassARRAY Nanodispenser in nanoliter volumes. SpectroCHIPs were analyzed with the Autoflex MALDI-TOF mass spectrometer (Brukar Daltonics, Hamburg, Germany) and the spectra were processed with the SpectroTYPER V4 software (SEQUENOM, San Diego, CA, USA).

#### 2.1.4. Haplogroup Determination

After successful genotyping of the mitochondrial DNA, the observed genotype results of the Pattanam excavated samples were compared with the revised Cambridge Reference Sequences (rCRS). Further, the genotype results of the Pattanam samples were compared with the worldwide data the ancestry of the samples was established based on the affiliated haplogroup of the samples (Table 1). Geospatial iso-frequency maps of West Eurasian-specific mitochondrial haplogroups were created using the kriging algorithm implemented in R package automap [12]. Haplogroup frequency reference data were compiled from published sources [13,14,15]. An ordinary kriging method for interpolation was used and we used R package ggplot2 for plotting the iso-frequency map [16].

#### 2.1.5. Radioactive AMS Dating of Ancient Samples

AMS radiocarbon dating of the Pattanam samples was done at β ANALYTIC INC., Miami, FL, USA. The raw material used for this purpose was the cremated bone carbonate and pre-treatment was bone carbonate extraction. Calibration was done with the 2013 database (INTLCAL13) from conventional radiocarbon dates and the results reported are 2-sigma calibrated (Appendix A). The modern reference standard used was 95% C14 activity oxalic acid (National Institute of Standards and Technology) calculated using the Libby 14C half-life (5568 years). The 13C/12C ratio was calculated relative to the PDB-1 standard. The conventional radiocarbon age represents the radiocarbon age corrected by isotopic fractionation and is calculated using Delta 13C. The conventional radiocarbon dates were used to get calendar calibrated results and are shown as 2-sigma calibrated results.

## 3. Results

### 3.1. Mitochondrial DNA Analysis

We have successfully generated good-quality genotype data from a total of 12 Pattanam ancient bone remains using the MassArray-based genotyping approach. After a careful comparison of mutations observed in the Pattanam samples and Phylotree, we determined the mitochondrial haplogroups of all 12 samples. We found that the mitochondrial haplogroup distribution of the Pattanam samples was highly heterogenous with the presence of both South Asian and West Eurasian-specific mtDNA haplogroups among all Pattanam samples. The haplogroups of the samples PT3, PT6, and PT8 were found to be M2a1a3, M6, and M3a1, respectively, which are South Asian mitochondrial haplogroups. In addition to this, haplogroups of samples PT10 and PT13 are South Asian-specific U1 and R5, respectively (Table 1). It is interesting to note that the mtDNA haplogroup U1 is present with low frequency in both Europe and India (mainly Kerala).

Many samples belonged to West Eurasian mitochondrial haplogroups whose prevalence in modern day South Asian populations is very low. Among these West Eurasian haplogroups, HV and subgroup HV4b were observed in samples PT1 and PT4, respectively. Haplogroup T and its subclade T1a9 were found in samples PT9 and PT2, respectively, while mtDNA haplogroup JT was present in sample PT3. The haplogroup of sample PT11 was uncharacterized because of the lack of resolution of genotyped mutation in this sample (Table 1).

Among the observed West Eurasian mtDNA haplogroups, haplogroup HV4b is mainly present in west Asia and Europe and absent in India [14]. Haplogroup T1a9 is present in Anatolia and Italy and is related with migration from near east to Europe [15]. Further, we made a comparison of the haplogroup prevalence of ancient Pattanam samples with the available frequency in worldwide populations. We are showing the frequency distribution of some of the major West Eurasian haplogroups viz, HV, T1a and JT on the map. Frequency of haplogroup HV is with its highest prevalence in the Caucasus and Middle East and Southern Europe (Figure 2) [13]. Its frequency is highest in Iraq (0.12) in the Middle East, followed by Armenia (0.07) and Azerbaijan (0.06) in the Caucasus [13]. Haplogroup T1a has the highest frequency in Balkan countries, Caucasus, and the Middle East and with lesser frequency in most of Europe (Figure 3 [17]. Its highest frequency is in Romania (0.08), Tunisia (0.07), and Iran (0.06). Haplogroup JT has the highest occurrence in Near East (0.45), the Alps (0.33), and Georgia (0.25) and is also prevalent in most of Europe (Figure 4 ) [13].

Among the South Asian haplogroups found in the Pattanam samples, M2a1 is present in most of the Indo-European and Dravidian tribes with coalescent ages of 7–9 KYA [18]. Haplogroup M3a1 is also mainly Indian-specific. It was mainly found among Indo–European and Dravidian tribes of North, Central, and South India [19]. It was also observed in the Kashmir region of North India [20]. Haplogroups M6 and R5 have widespread occurrences in India [19]. Mitochondrial haplogroup M6 is highly prevalent in Indus valley and Eastern coastal region of India towards the Bay of Bengal. Haplogroup R5 is the most ancient and second frequent sub-haplogroup of R. This haplogroup is most frequent among Indian caste groups compared to tribes and absent among Austroasiatic speakers.

### 3.2. AMS Radiocarbon Age Estimates

We estimated the AMS radiocarbon ages of three specimens, namely PT3, PT4, and PT5, using cremated bone carbonate. The observed 13C/12C ratio for sample PT3 was −24.0 and the measured radiocarbon age was 2080 ± 30 BP. While for the sample the PT4 ratio was −8.7 and the measured radiocarbon age was 1990 ± 30 BP. The observed ratio (13C/12C) for sample PT5 was −9.2, with the measured radiocarbon age being 2130 ± 30 BP. The 2-sigma calibrated result with 95% probability for samples PT3, PT4, and PT5 was Cal BC 200-45, Cal BC 395-35, and Cal BC 540-395, respectively.

## 4. Discussion

South Asia was one of the most diverse regions of the world with the mighty Indian Ocean serving as the center of attraction for sailors, merchants, and adventures. South Asia presents a unique set of genetic differentiation and extensive structuring at its roots due to a complex evolutionary antiquity and endogamous practices within different population groups. The genetic differentiation is supported by the linguistic differences among these population groups. India is a major part of the South Asian horizon, the social history, cultural practices, migration, trade practices, religious migration, and agricultural movements have made a significant effect on the genetic structuring of India. Ports played an important role in the second phase of human urbanization during the 500 BCE to 500 CE phase with maritime networks across continents and oceans. In this period, three ports played as trade emporia. In their chronological order, the first one was Barygaza in the modern Gujarat coast, then Muziris in the Kerala coast, and Tamralipti in the Bengal coast.

Of these, Pattanam or Muziris have a special significance because of the rise of Greco Roman cultures and the establishment of the Roman empire that coincides with the peak phase of the Pattanam site that is 100 BCE to 300 CE [1]. Already, the material evidence at Pattanam vouch for the presence of at least three dozen of linguistic groups from the Mediterranean to South China [1]. During this time, three trade systems, namely the silk maritime road from South China, the spice maritime road from South India, and the incense maritime roads from South Arabia transformed the Indian Ocean into a trade lake. When these three maritime roads joined with the Red Sea and Mediterranean maritime roads, there was an intensity of commercial and cultural exchanges. Numerous linguistic communities from different parts of the Greco-Roman world to eastern Mediterranean, western and eastern Africa to South Arabia to North western India to Coromandel India to Burmese, Malayan, and south China coastal regions moved along with the goods and technologies exchanged through the various maritime roads.

Several genomic studies that were published earlier [4,5,6,21,22,23] support the migration of many groups in and from India. Footprints of all these earlier migrations are present either in the form of the modern Indian gene pool or in the form of excavated ancient remains of the people of those ancient times. Deciphering the genetic history of the latter using the blooming ancient DNA field is an exciting field of interest. Harsh geographical and climatic conditions make the recovery of ancient DNA a challenging task in India but not an impossible one [5,24,25].

An in-depth analysis of skeletal remains excavated from the Pattanam archaeological site supports the claim and in the current study, the first attempt at genotyping of mtDNA markers help to understand the demographic distribution of the Pattanam location [6,9]. The results show a mixture of ethnicity from West Eurasian ancestry to South Asian ancestry. The frequency distribution of haplogroups reflects a high occurrence of West Eurasian (T, JT, and HV) and South Asian-specific mitochondrial haplogroups (M2a, M3a, R5, and M6). The presence of haplogroups HV, T1a9, JT, HV4b, T, and U1 strongly supports the presence of West Eurasian and European ancestry, whereas M2a1a3, M6, M3a1, and R5 haplogroups support the South Asian ancestry. Some of these West Eurasian mtDNA haplogroups such as T1a9 and HV4b have been found exclusively in European and West Asian populations and are absent in Indian populations. The presence of such a diverse set of haplogroups reflects the mix of several cultural and religious groups that must have immigrated, settled, and eventually died out on the south-west coast of India. The AMS radiocarbon dates of three specimens (PT3, PT4, and PT5) range from 540 to 45 BCE., which strongly agrees with earlier archaeological observations with artifacts found at the Pattanam site. The excavations at this site between 2007 and 2022 indicate the presence of a multicultural port complex in the early historical phase (300 BCE–500 CE) [1].

## 5. Conclusions

Our present study sheds light on the genetic makeup of ancient human populations at the Pattanam archaeological site and reinforces the early historical occupation of culturally, religiously, and ethnically diverse groups at the Pattanam archaeological site. According to literary sources from centuries before the present time as well as based on ancient Greek and Roman sources, the evidence of cultural and commercial goods clearly connects the indigenous worlds of the Greco-Romans, the Egyptians, and the West Asians. Through several rounds of archaeological excavations and expeditions at the archaeological site of Pattanam, intensive maritime exchanges have been established between the eastern Mediterranean and ancient South India since 2006. The extensive use of Indian and West Asian precious and semi-precious stones in the various provinces of the Empire in Greece, Syria, Egypt, but especially Rome and Italy, indicates a great possibility of cultural assimilation. Images of a leaping lion and Fortuna (Greek-Roman goddess of fortune known in Rome and having a symbolic meaning) from the eighth century BC were excavated along with semi-precious stones from the archaeological site of Pattanam. The current genetic study of skeletal remains at the Pattanam Archaeological Site is the first of its kind, providing very good confirmation of these archaeological finds. The genetic evidence of many mtDNA haplogroups specific to either the Middle East or Europe among the ancient individuals of the Pattanam site further connects the threads of evidence derived from the material culture and artifacts. This further proves that Pattanam was not only a place for trade and material exchange, but also a possible settlement site for human populations involved in these trade affairs. This early historical demographic settlement included both local populations and the influx of diverse populations from the Middle East and Europe. Furthermore, the high diversity observed among both South Asian and West Eurasian mtDNA haplogroups in ancient Pattanam individuals supports the diverse nature of these settled human populations. In conclusion, our findings are consistent with archaeological observations, demonstrating the presence of multicultural groups who came to the port city of Pattanam mainly as traders. Our current study is the first attempt to genetically structure the demographic classification of the Pattanam archaeological site and to deduce its maternal ancestry.

## Figures and Tables

**Figure 1 genes-14-00963-f001:**
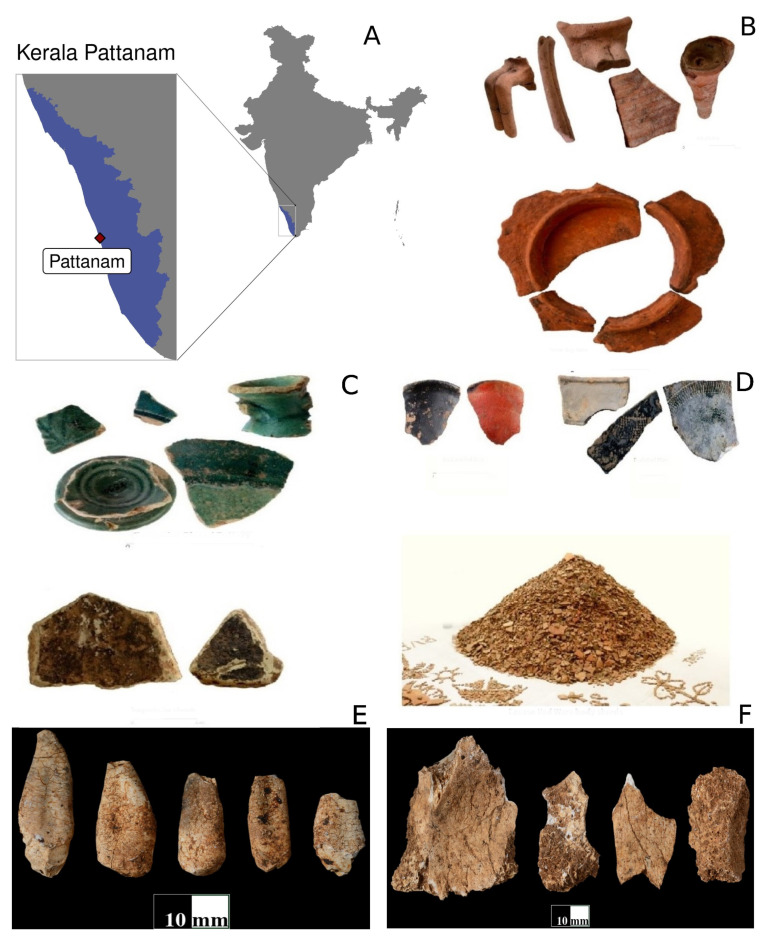
(**A**) The location of the Pattanam site in Kerala, India. (**B**) The Mediterranean (Roman) pottery, represented in the Amphora and Terra Sigilletta sherds. (**C**) West Asian and South Arabian ceramics include turquoise glazed pottery, torpedo jars, and ovoid jar fragments. (**D**) Indian ceramics include the black and red ware (BRW), red course ware, and Indian rouletted ware (IRW). (**E**,**F**) Human skeletal remains (specimen IDs REG.NO.336 and REG.NO.338 respectively) recovered from excavations at trench TRENCH-XLII-035-06 of Pattanam site.

**Figure 2 genes-14-00963-f002:**
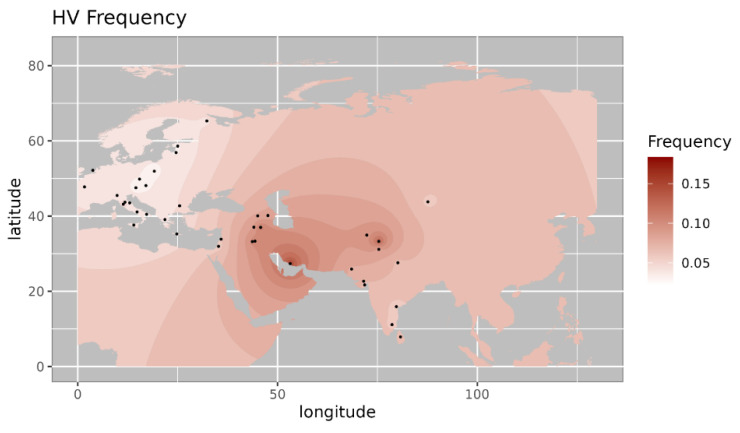
Iso-frequency map of mitochondrial DNA haplogroup HV inferred from the worldwide frequency distribution. Sampling locations are represented by black dots in the plotting area. The geospatial pattern indicates that HV is highly prevalent in the near East, with minor occurrence in Northwest India.

**Figure 3 genes-14-00963-f003:**
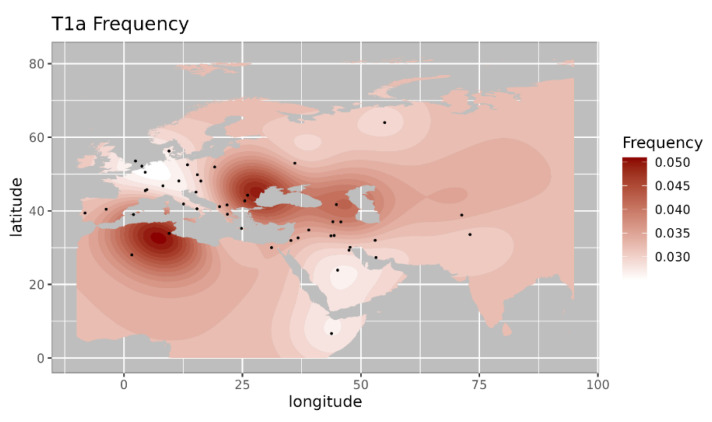
Iso-frequency map of mitochondrial DNA haplogroup T1a inferred from the worldwide frequency distribution. Sampling locations are represented by black dots in the plotting area. The geospatial pattern indicates that T1a is highly prevalent in Balkan, Caucasus, and Southern Europe, with minor occurrence in the near East.

**Figure 4 genes-14-00963-f004:**
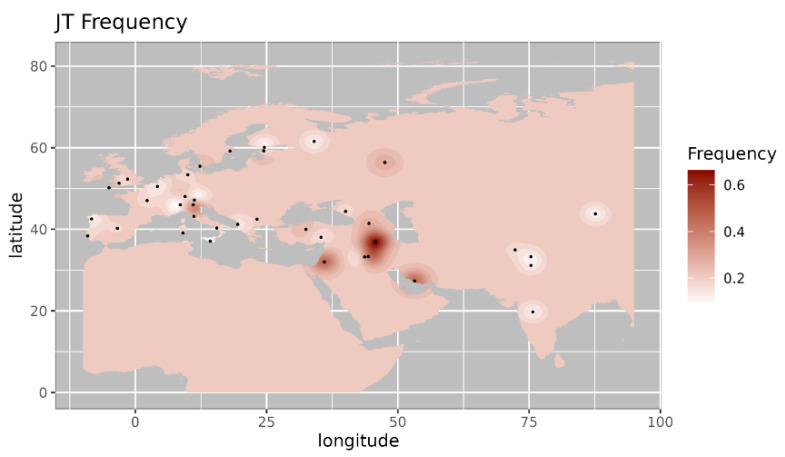
Iso-frequency map of mitochondrial DNA haplogroup JT inferred from the worldwide frequency distribution. Sampling locations are represented by black dots in the plotting area. The geospatial pattern indicates that JT is highly prevalent in the Caucasus and Southern Europe, with minor occurrence in the near East.

**Table 1 genes-14-00963-t001:** The ancestry of 12 samples was established based on haplogroup defining mtDNA mutations.

Sample Code	KCHR Sample Reg. No.	Observed Haplogroup	Ethnicity/Haplogroup Distribution
PT1	PT07 II BON 001	HV	West Eurasian haplogroup found throughout West Asia and
			south-eastern Europe
PT2	PT07 I BON004	T1a9	Haplogroup T is found in native Europeans with high concentrations
			At the eastern Baltic Sea.
PT3	PT07 I BON 005	JT	West Eurasian haplogroup and it is extremely common among
			ancient Etruscans
PT4	PT 07 I BON015	HV4b	West Eurasian haplogroup found throughout West Asia and
			south-eastern Europe
PT5	PT 07II BON 028	M2a1a3	Very common in South Asia and Ancient Indian specific haplogroup
PT6	PT 07I BON 010	M6	Found mainly in South Asia and India
			(With a high concentration in Kashmir, mid-eastern India, and Kerala).
PT7	PT 07III BON 032	U	Found in Turkmenistan, Northwest Caucasian, and observed
			in Eastern Europeans
PT8	PT09 XII BON 012	M3a1	Ancient India-specific and found in South Asia.
PT9	PT10 XVI & PT XVII BON 042	T	Found in Turkmenistan, Northwest Caucasian, and observed in Eastern Europeans
PT10	PT 11 XXIX TEE 222	U1	Mostly found in European and found in Kerala Jews
			and Jews from other parts of the world
PT11	PT13 XLII BON 317	UC	No mutation was observed
PT12	PT13 XLII BON 342	R5	Widely spread in Indian sub-continent.

## Data Availability

No new sequence data were generated. Not applicable.

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
