# Peer review of "Deciphering the West Eurasian Genetic Footprints in Ancient South India"

_genes, 2023, doi:10.3390/genes14050963_

Round 1

Reviewer 1 Report

This study reports on the potential migration of the Western Eurasian population into a coastal town in Southern India.  Using ancient DNA technology, mitochondrial DNA from 12 ancient samples have been amplified and genotyped.  Through identifying the mitochondrial haplogroups, these individuals were found to belong to European, middle eastern, and Indian populations.  Overall, this is an interesting study that adds to the literature related to ancient human migrations and hence the findings are important for anthropological and human genetic research.  The manuscript was written well and easy to follow.  My only minor comment is that the discussion could have been a bit more elaborate.  The authors could throw some speculations based on the historical narratives available about who those migrated people could be and what could be the cause for the migration (trade, escape from religious persecution, seeking better lifestyles, etc).

Reviewer 2 Report

Thank you for the opportunity to review the paper entitled “Deciphering the West Eurasian Genetic Footprints in Ancient South India.”  I have the following comments:

1.       In the Materials and methods section you mention that bone remaining from cremated remains were used to extract DNA from.  Could you please elaborate on this? How many actual bone samples were available vs. teeth that remained? Also, which part of the skeleton / which bones were used for extraction?

2.       Line 133: The sentence is unclear as to how the primar pools were set up; were the primers split into four pools each containing the 94 primers? Or were the primers split into 4 pools each with different primers.  Just clarify, so readers that are not familiar with the process may better understand.

3.       Line 142: The haplogroup determination paragraph should form part of the result section.

4.       Line 150: The results from the radiocarbon dating should also form part of the results and can be explained in more detail, giving an explanation of the results.

5.       Line 164: Are you referring to modern day prevalence in South Asia? Just clarify the sentence, because it seems that the prevalence may have been higher during the specified time period of the samples used in this study. 

6.       Are the isofrequency maps based solely on the data from other studies? It appears so… If indeed so, I would suggest adding your samples to the figures so one can compare it visually.  Also, all three figures should follow the same format – Fig. 3 is represented differently, which disrupts that flow of the paper. 

7.       The comparison between your samples and other samples indicated on the figures is not well explained in the discussion.  The discussion needs much elaboration – you have so much info; genetics, archaeology, radiocarbon dating, etc… Yet the discussion does not tell us anything about the populations that lived during the specified time period.  The discussion is thus wholly incomplete.

Extensive language editing is also recommended. 

Round 2

Reviewer 2 Report

Thank you for the opportunity to re-review the article. 
The authors has made the suggestions given on the first review, and the discussion has been much improved.

I would gladly see this work published.